# On the choice of calibration metrics for "high flow" estimation using hydrologic models

Naoki Mizukami[1], Oldrich Rakovec[2,3], Andrew J. Newman[1], Martyn P. Clark[4], Andrew W. Wood[1], Hoshin V. Gupta[5], and Rohini Kumar[2]

[1]National Center For Atmospheric Research, Boulder CO, USA
[2]UFZ-Helmholtz Centre for Environmental Research, Leipzig, Germany
[3]Czech University of Life Sciences, Prague, Czech Republic
[4]Coldwater Laboratory, university of Saskatchewan, Canmore, Alberta, Canada
[5]Department of Hydrology and Atmospheric Sciences, The University of Arizona, Tucson, Arizona, USA

**Correspondence:** Naoki Mizukami (mizukami@ucar.edu)

**Abstract.** Calibration is an essential step for improving the accuracy of simulations generated using hydrologic models. A key modeler decision is selecting the performance metric to be optimized. It has been common to use squared error performance metrics, or normalized variants such as Nash-Sutcliffe Efficiency (NSE), based on the idea that their squared-error nature will emphasize the estimates of high flows. However, we conclude that NSE-based model calibrations actually result in *poor* reproduction of high flow events, such as the annual peak flows that are used for flood frequency estimation. Using three different types of performance metrics, we calibrate two hydrological models at daily step, the "Variable Infiltration Capacity" model (VIC) and the "mesoscale Hydrologic Model" (mHM) and evaluate their ability to simulate high flow events for 492 basins throughout the contiguous United States. The metrics investigated are (1) NSE, (2) Kling-Gupta Efficiency (KGE) and its variants, and (3) Annual Peak Flow Bias (APFB), where the latter is an application-specific metric that focuses on annual peak flows. As expected, the APFB metric produces the best annual peak flow estimates; however, performance on other high flow related metrics is poor. In contrast, the use of NSE results in annual peak flow estimates that are more than 20% worse, primarily due to the tendency of NSE to underestimate observed flow variability. On the other hand, the use of KGE results in annual peak flow estimates that are better than from NSE owing to improved flow time series metrics (mean and variance), with only a slight degradation in performance with respect to other related metrics, particularly when a non-standard weighting of the components of KGE is used. Stochastically generated ensemble simulations based on model residuals show the ability to improve the high flow metrics, regardless of the deterministic performances. However, we emphasize that improving the fidelity of streamflow dynamics from deterministically calibrated models is still important as it may improve high flow metrics (for the right reasons). Overall, this work highlights the need for a deeper understanding of performance metric behavior and design in relation to the desired goals of model calibration.

# 1 Introduction

Computer-based hydrologic, land-surface, and water balance models are used extensively to generate continuous long-term hydrologic simulations in support of water resources management, planning and decision making. Such models contain many empirical parameters that cannot be estimated directly from available observations, hence the need for parameter inference by
means of the indirect procedure known as calibration (Gupta et al., 2006). In general, all such models require some degree of calibration to maximize their ability to adequately reproduce the observed dynamics of the system response (e.g., streamflow).

A key decision in model calibration is the choice of performance metric (also known as "objective function") that measures the goodness of fit between the model simulation and system observations. The performance metric can substantially affect the quality of the calibrated model simulations. The most widely used performance metrics are based on comparisons of simulated
and observed response time series, including the Mean Squared Error (MSE), Nash-Sutcliffe Efficiency (NSE; a normalized version of MSE) and Root Mean Squared Error (RMSE; a transformation of MSE). Many previous studies have examined different variants of these metrics (e.g., see Oudin et al., 2006; Kumar et al., 2010; Pushpalatha et al., 2012; Price et al., 2012; Wöhling et al., 2013; Ding et al., 2016; Garcia et al., 2017), including their application to transformations of the system response time series to emphasize performance for specific flow regimes (e.g. use of logarithmic transformation to target low
flows), or using combinations of different metrics to obtain balanced performance on different flow regimes.

As an alternative to metrics that measure the distance between response time series, the class of '*hydrologic signature*' metrics (e.g., Olden and Poff, 2003; Shamir et al., 2005; Gupta et al., 2008; Yilmaz et al., 2008; Westerberg and McMillan, 2015; Westerberg et al., 2016; Addor et al., 2017a) has been gaining popularity for hydrologic model calibration (Yadav et al., 2007; Westerberg et al., 2011; Shafii and Tolson, 2015; Kavetski et al., 2018). A hydrologic signature is a metric that quantifies
a targeted property or behavior of a hydrologic time series (e.g., that of a specific portion such as peaks, recessions, water balance, flow variability, flow correlation structure, etc.), in such a way that it is informative regarding a specific hydrologic process of a catchment (Yilmaz et al., 2008).

The use of hydrologic signatures to form metrics for model calibration requires selecting a full set of appropriate signature properties that are relevant to all of the aspects of system behavior that are of interest in a given situation. As discussed by Gupta
et al. (2008), the use of multiple hydrologic signatures for model calibration involves the use of multi-objective optimization (Gupta et al., 1998) in which a trade-off among the ability to optimize different signature metrics must be resolved. This means that, in the face of model structural errors, it is typically impossible to simultaneously obtain optimal performance on all of the metrics (in addition to the practical difficulty of determining the position of the high dimensional Pareto front). Further, if only a small subset of signature metrics is used for calibration, the model performance in terms of the non-included metrics
can suffer (Shafii and Tolson, 2015). The result of calibration using a multi-objective approach is a Pareto-set of parameters, where different locations in the set emphasize different degrees of fit to the different hydrological signatures.

In general, water resources planners focus on achieving maximum accuracy in terms of specific hydrologic properties and will therefore select metrics that target the requirements of their specific application while accepting (if necessary) reduced model skill in other aspects. For example, in climate change impact assessment studies, reproduction of monthly or seasonal

streamflow is typically more important than behaviors at finer temporal resolutions, and so hydrologists typically use monthly rather than daily error metrics (Elsner et al., 2010, 2014). Hereafter this metric is referred to as 'application specific metric'. It is worth-noting that the application specific metric can be a hydrologic signature metric. For example, high flow volume based on the flow duration curve characterizes the surface flow processes and may be interest for estimation of flood frequency.

In this study, we examine how the formulation of the performance metric used for model calibration affects the overall functioning of system response behaviors generated by hydrologic models, with a particular focus on high flow characteristics. The specific research questions addressed in this paper are:

1. How do commonly used time-series based performance metrics perform compared to the use of an application specific metric?

2. To what degree does use of an application specific metric result in reduced model skill in terms of other metrics not directly used for model calibration?

We address these questions by studying the high flow characteristics and flood frequency estimates for a diverse range of 492 catchments across the Contiguous United States (CONUS) generated by two models: the mesoscale Hydrologic Model (mHM; Samaniego et al., 2010; Kumar et al., 2013b) and the Variable Infiltration Capacity (VIC; Liang et al., 1994) model. Our focus on high flow estimation is motivated by: (a) their importance to a wide range of hydrologic applications related to high flow characteristics (e.g., flood forecasting, flood frequency analysis), their relevance to historical change and future projections (Wobus et al., 2017); and (b) persistent lack of community-wide awareness of the pitfalls associated with use of squared error type metrics for high flow estimation. Specifically, we compared and contrasted the model simulation results of the calibration based on metric (1) NSE, (2) Kling-Gupta Efficiency (KGE) and its variants, and (3) Annual Peak Flow Bias (APFB) – with focus on understanding and evaluating the appropriateness of different metrics to capture observed high flow behaviors across a diverse range of U.S. basins. We also discuss the implications of the choice of different calibration metrics based on stochastic ensemble simulations generated based on remaining model residuals.

The remainder of this paper is organized as follows. Section 2 shows how the use of NSE for model calibration is counter-intuitively problematic when focusing on for high flow estimation. This part of the study is motivated by our experience with CONUS-wide annual peak flow estimates and serves to motivate the need for our large-sample study (Gupta et al., 2014). Section 3 describes the data, models and calibration strategy adopted. Section 4 then presents the results followed by discussion in Section 5. Concluding remarks are provided in Section 6.

## 2 Motivation

[9] One of the earliest development of a metric used for model development is by Nash and Sutcliffe (1970), who proposed assessing MSE relative to the observation mean; Nash Sutcliffe Efficiency (NSE). A key motivation was to quantify how well the updated model outputs performed when compared against a simple benchmark (the observation mean). Since then, such squared error metrics have been predominantly used for model evaluation as well as for model calibration. Furthermore,

MSE-based metrics have been thought to be useful in model calibration to reduce simulation errors associated with high flow values, because these metrics typically magnify the errors in higher flows more than in the lower flows due to the fact that the errors tend to be heteroscedastic. Although Gupta et al. (2009) showed theoretically how and why the use of NSE and other MSE-based metrics for calibration results in the underestimation of peak flow events, our experience indicates that this notion continues to persist almost a decade later (Price et al., 2012; Ding et al., 2016; Seiller et al., 2017; de Boer-Euser et al., 2017). Via an algebraic decomposition of the NSE into 'mean error', 'variability error', and 'correlation' terms, Gupta et al. (2009) demonstrate that use of NSE for calibration will underestimate the response variability by a proportion equal to the achievable correlation between the simulated and observed responses; i.e., the only situation in which variability is not underestimated is the ideal but unachievable one when the correlation is 1.0. They further show that the consequence is a tendency to underestimate high flows while overestimating low flows (see Fig.3 in Gupta et al., 2009).

Our recent large sample calibration study (Mizukami et al., 2017) made us strongly aware of the practical implications of this problem associated with the use of NSE for model calibration. Figure 1 illustrates the bias in the model's ability to reproduce high flows when calibrated with NSE. The plot shows distributions of annual peak flow bias at 492 Hydro-Climate Data Network (HCDN) basins across the CONUS for the VIC model using with three different parameter sets determined by Mizukami et al. (2017). Note that the collated parameter set is a patchwork quilt of partially calibrated parameter sets, while the other two sets were obtained via calibration with NSE using the observed data at each basin. The results clearly demonstrate the strong tendency to underestimate annual peak flows at the vast majority of the basins (although calibration at individual basins results in less severe underestimation than the other cases). Figures 1 (b-d) clearly show that annual peak bias is strongly related to variability error, but not to mean error (i.e., water balance error). Even though the calibrations resulted in statistically unbiased results over the sample of basins, there is a strong tendency to severely underestimate annual peak flow due to fact that NSE results in poor statistical simulation of variability. Clearly, the use of NSE-like metrics for model calibration is problematic for the estimation of high flows and extremes. However, improving only simulated flow variability may not improve high flow estimates in time. It likely also requires improvement of the mean state and daily correlation.

In general, it is impossible to improve the simulation of flow variability (to improve high flow estimates) without simultaneously affecting the mean and correlation properties of the simulation. To provide a way to achieve balanced improvement of simulated mean flow, flow variability, and daily correlation, Gupta et al. (2009) proposed the Kling-Gupta Efficiency (KGE) as a weighted combination of the three components that appear in the theoretical NSE decomposition formula, and showed that this formulation improves flow variability estimates. KGE is expressed as:

$$KGE = 1 - \sqrt{[S_r(r-1)]^2 + [S_\alpha(\alpha-1)]^2 + [S_\beta(\beta-1)]^2} \qquad \alpha = \frac{\sigma_s}{\sigma_o}, \beta = \frac{\mu_s}{\mu_o} \qquad (1)$$

where $S_r$, $S_\alpha$ and $S_\beta$ are user specified scaling factors for the correlation (r), variability ratio ($\alpha$), and mean ratio ($\beta$) terms; $\sigma_s$ and $\sigma_o$ are the standard deviation values for the simulated and observed responses respectively, and $\mu_s$ and $\mu_o$ are the corresponding mean values. In a balanced formulation, $S_r$, $S_\alpha$ and $S_\beta$ are all set to 1.0. By changing the relative sizes of the $S_r$, $S_\alpha$ or $S_\beta$ weights, the calibration can be altered to more strongly emphasize the reproduction of flow timing, statistical variability, or long-term water balance.

The results of the Mizukami et al. (2017) large sample study motivated us to carry out further experiments to investigate how the choice of performance metric affects the estimation of peak and high flow. Here, we examine the extent to which altering the scale factors in KGE can result in improved high flow simulations compared to NSE. We also examine the results provided by use of an application specific metric, here taken as the %bias in annual peak flows.

## 3  Datasets, Methods and Methods

We use two hydrologic models; VIC and mHM. The VIC model, which includes explicit soil-vegetation-snow processes, has been used for a wide range of hydrologic applications, and has recently been evaluated in large-sample predictability benchmark study (Newman et al., 2017). The mHM model has been shown to provide robust hydrologic simulations over both Europe and the US (Kumar et al., 2013a; Rakovec et al., 2016b) and is currently being used in application studies (e.g., Thober et al.,
2018; Samaniego et al., 2018). We use observed streamflow data at the HCDN basins and daily basin meteorological data from Maurer et al. (2002) for the period from 1980 through 2008, as compiled by the CONUS large sample basin dataset over a wide range of climate regimes (Newman et al., 2014; Addor et al., 2017b). The use of the large sample dataset is recommended to obtain general and statistically robust conclusions (Gupta et al., 2014). In the context of flood mechanisms across CONUS, large flood events are due to precipitation excess in conjunction with antecedent soil moisture states at the majority of the catchments
except that rapid snowmelt events are primarily responsible for floods over the mountainous West (Berghuijs et al., 2016). Both models are run at a daily time step, and each of model is calibrated separately for each of the 492 study basins (see Fig. 1a for the basin locations) using several different performance metrics. Although sub-daily simulation is preferable for some flood events, such as flash floods, the effects of the performance metrics on the calibrated high flow estimates are independent of the simulation time step. Furthermore, instantaneous peak flow (at sub-daily scale) is strongly correlated with daily mean flows
(Dieter and Arns, 2003; Ding et al., 2016), justifying that daily simulations still provide useful information for instantaneous peak flow estimates. We use a split-sample approach (Klemes, 1986) for the model evaluation. The hydrometeorological data is split into a calibration period (October 1, 1999 - September 30, 2008) and an evaluation period (October 1, 1989 - September 30, 1999), with a prior 10-year warm-up when computing the statistics for each period.

The model parameters calibrated for each model are the same as previously discussed: VIC (Newman et al., 2017; Mizukami
et al., 2017) and mHM (Rakovec et al., 2016a, b). Although alternative calibration parameter sets have also been used by others, particularly for VIC (Newman et al., 2017), the purpose of this study is purely to examine the effects of performance metrics used for calibration, and not to obtain "optimal" parameter sets. Each model is identically configured for each of the 492 basins. Both models use the same set of underlying physiographical and meteorological datasets, so that performance differences can be attributed mainly to the strategy used to obtain the parameter estimates.
Optimization is performed using the Dynamically Dimension Search (DDS, Tolson and Shoemaker, 2007) algorithm. Five performance metrics are used for the calibration/evaluation purpose: 1) KGE, 2) KGE-$2\alpha$, 3) KGE-$5\alpha$, and 4) APFB (Annual Peak flow bias) and 5) NSE. The first three metrics are KGEs with different scaling factor combinations ($S_r$, $S_\alpha$ and $S_\beta$) = (1,1,1), (1,2,1), and (1,5,1) in Eq.(1), respectively; because variability is strongly correlated with annual peak-flow error (see

Fig. 1 c), we explore the impact of rescaling the variability error term in Eq. 1. The forth metric, APFB, is our application-specific high flow metric, defined as:

$$APFB = \sqrt{[(\mu_{peakQ_s}/\mu_{peakQ_o} - 1)]^2} \qquad (2)$$

where $\mu_{peakQ_s}$ is the mean of the simulated annual peak flow series and $\mu_{peakQ_O}$ is the mean of the observed annual peak flow series. Finally, we took NSE as a benchmark performance metric, and compared and contrasted the simulations based on other performance metrics.

The most common choice of KGE scaling factor for hydrologic model calibration has been to set all of them to unity. We applied the KGE in different variants (i.e., with non-unity scaling factors) which to best of our knowledge have not been studied so-far. Note that this scaling is only used to define the performance metric used in model calibration; all performance evaluation results shown in this paper use KGE computed with $S_r$, $S_\alpha$ and $S_\beta$ all set to 1.0.

## 4 Results

### 4.1 Overall Simulation Performance

First, we focus on the general overall performance for the daily streamflow simulations as measured by the performance metrics used. Figures 2 and 3 show the cumulative distributions of the model skill during the evaluation period across the 492 catchments in terms of KGE and its three components: (a) $\alpha$ (standard deviation ratio), (b) $\beta$ (mean ratio), (c) r (linear correlation) for VIC (Fig. 2) and mHM (Fig. 3). Considering first the result obtained using KGE, both models, at the median values of the distributions, show improvement in the variability error by approximately 20% over that obtained using the NSE-based calibration (Figs. 2a and 3a). The plots, however, indicate a continued statistical tendency to underestimate observed flow variability even when the (1,5,1) component weighting is used in the scaled KGE-based metric. The corresponding median $\alpha$ and r values obtained for KGE are: $(\alpha, r) = (0.83, 0.74)$ for VIC and $(\alpha, r) = (0.94, 0.82)$ for mHM. Interestingly, the VIC results are more sensitive than mHM to variations in the $S_\alpha$ weighting. For VIC, the variability estimate continues to improve with increasing $S_\alpha$ (median moves closer to 1.0), but simultaneously leads to overestimation of the mean values ($\beta$) and deterioration of correlation (r).

The use of APFB as a calibration metric yields poorer performance for both models, on all of the individual KGE components (wider distributions for $\alpha$ and $\beta$, and distribution of r shifted to the left), and consequently on the overall KGE value as well (distribution shifted to the left). In terms of performance as measured by NSE, the use of KGE with the original scaling factors ($\alpha = 1$) results in 3-10% lower NSE than those obtained with the NSE-based calibration case (plots not shown). This is consistent with the expectation that an improvement in the variability error ($\alpha$ closer to unity) leads to deterioration in the NSE score. In general, all the calibration results from both models are consistent with the NSE-based calibration characteristics described in Gupta et al. (2009).

## 4.2 High flow simulation performance

Next, we focus on the specific performance of the models in terms of simulation of high flows. As expected, use of the application-specific APFB metric (Eq. 2) leads to the best estimation of annual peak flows for both models (Figure 4 a and b), while use of NSE produces the worst peak flow estimates. Simply switching from NSE to KGE improves APFB by approximately 5% for VIC and 10% for mHM at the median value during evaluation period. Improvement of APFB occur at over 85% of 492 basins for both models. Note that the inter-quartile range of the bias across the basins becomes larger for the evaluation period compared to the calibration period. This is even more pronounced when APFB is used as the objective function (see the results from mHM; Figure 4 a and b), indicating that the application specific objective function results in overfitting, and consequently the model is less transferable in time than when the other metrics are used for calibration.

Figure 4c and d show the high flow simulation performance in terms of another high flow related metric - the percent bias in the runoff volume above the 80th percentile of the daily flow duration curve (FHV; Yilmaz et al., 2008). Interestingly, FHV is not reproduced better by the APFB calibrations compared to the other objective functions, particularly for VIC. The implication is that, in this case, the application specific metric only provides better results for the targeted flow characteristic (here the annual peak flow), but can result in poorer performance for other flow properties (even the closely related annual peak flow). While the mHM model calibrated with APFB does produce a nearly unbiased FHV estimate across the CONUS basins, the inter-quartile range is much larger than that obtained using the other calibration metrics. The VIC model based results also exhibit larger variability in the FHV bias across the study basins.

## 4.3 Implication for flood frequency estimation

Annual peak flow estimates are generally used directly in the flood frequency analysis. Figure 5 shows estimated daily flood magnitudes at three return periods (5-, 10-, 20-yr) using the five different sets of calibration results. Although many practical applications (e.g., floodplain mapping and water infrastructure designs) require estimates of higher extreme events, we focus on 20-yr (0.95 exceedance probability) for the highest extremes, given use of only 20-years of data for this study; this is to avoid the need for extrapolation of extreme events via theoretical distribution fitting. For this evaluation case (of annual flood magnitudes), we use the combined calibration and evaluation periods.

Figure 5 shows results that are consistent with Figure 4, although more outlier basins are found to exist for estimates of flood magnitude at the three return periods. The KGE-based calibration improves flood magnitude estimates (compared to NSE) at all three return periods for both models. In particular, mHM especially exhibits a clear reduction of the bias by 10% at the median compared to the NSE calibration case. The APFB calibration further reduces the bias by 20% and 10% for VIC and mHM respectively. However, regardless of the calibration metric, for both models the peak flows at all return periods are underestimated; although mHM underestimates the flood magnitudes to a lesser degree due to its smaller underestimation of annual peak flow estimates. Even though APFB is less than 5% at the median value for mHM calibrated with APFB (Figure 4), the 20-yr flood magnitude is underestimated by almost 20% at the median (Figure 5). Also, the degree of underestimation of flood magnitude becomes larger with longer return periods.

## 5   Discussion

While both models show fairly similar trends in skill for each performance metric, it is clear from our large sample study of 492 basins that the absolute performance of VIC is inferior to that of mHM, irrespective of choice of evaluation metric. A full investigation on to why VIC does not perform at the same level of mHM is clearly of interest, but is left for future work. To improve the performance of VIC it may be necessary to perform rigorous sensitivity tests similar to comprehensive sensitivity studies that have included investigation of hard-coded parameters in other more complex models (e.g., Mendoza et al., 2015; Cuntz et al., 2016). Below, we discuss our results in the context of usage of different performance metrics, in regard to remaining aspects of model errors, and provide suggestions for potential improvement of the high flow simulations related metrics.

### 5.1   Consideration of application specific metric

Although the annual peak flow estimates improve by switching calibration metrics from NSE to KGE and KGE to APFB, the flood magnitudes are underestimated at all of the return periods examined no matter which performance metric is used for calibration. While the APFB calibration improves, on average, the error of annual peak flow over the 20-year period, the flood magnitude estimates at several percentile or exceedance probability levels are based on estimated peak flow series. Therefore, improving only the bias does not guarantee accuracy of the flood magnitudes at a given return period. Following Gupta et al. (2009), events that are more extreme may be affected more severely by variability errors when examining the series of annual peak flows, particularly because this performance metric accounts only for annual peak flow bias. Figure 6 shows how the estimates of flood magnitudes at the 20-yr return period (top panels) and 5-yr return period (bottom panels) are related to variability error and bias of annual peak flow estimates. As expected, the more extreme (20-yr return period) flood estimates are more strongly correlated with estimates of the variability of annual peak flows than with the 20-yr bias of the annual peak flow series. For the less extreme (5-yr return period) events, this trend is flipped and flood magnitude errors are more correlated with the bias.

### 5.2   Consideration of model residuals

The calibrated models do improve the flow metrics including both time series metrics (mean, variability, etc.) and/or application specific metrics, depending on the performance metrics used for the calibration. However, residuals always remain after the model calibration because the model never reproduces the observations perfectly. Recently, Farmer and Vogel (2016) discussed the effects of neglecting residuals on estimates of flow metrics, particularly errors in statistical moments of flow time-series (mean, variance, skewness and so on). In the context of this study for the high flow simulations, lets focus on the flow variability (i.e., variance) component for observation and model simulations, which can expressed by the following equation:

$$\text{Var}(o) = \text{Var}(s + \epsilon) = \text{Var}(s) + \text{Var}(\epsilon) + 2\text{COV}(s, \epsilon) \tag{3}$$

where, Var(X) is variance of X, and COV(X,Y) is covariance between X and Y, $o$ is the observed time series, $s$ is simulated time series from calibrated model and $\epsilon$ the residuals. The observation time series can be expressed as the sum of the model simulation and residual terms (denoted as $\hat{s} = s + \epsilon$). As seen in Eq. 3, neglecting the residuals can match the observed variability, only when the variance of the residuals is offset by covariance between the simulation and residuals i.e., COV($s,\epsilon$). Of course, this condition is not fulfilled (in real word simulation studies). In our calibration results (as discussed above), the observed flow variability is underestimated for both models in the majority of the study basins for nearly all performance metrics used for the calibration (Figure 2a and 3a).

To gain more insight into this topic, we examine how stochastically generated residuals, once re-introduced to the simulated flows, can affect the performance metrics. We consider three performance metrics for this analysis: NSE, KGE, and APFB. Figure 7 shows the distributions of flow residuals produced by the calibrated models. The APFB calibration that produces the worst temporal pattern of flow time series (the lowest correlation shown in Figure 2d and 3d) produces wider residual distributions. Following the method of Bourgin et al. (2015); Farmer and Vogel (2016), 100 sets of synthetic residual time series ($\epsilon$) are stochastically generated by sampling the residuals of the calibrated flow (i.e., simulation during the calibration period) for each model and added to the respective modeled flow during the evaluation period. The method randomly samples the residuals from the residual pool based on the flow magnitude. For each of the 100 residual amended flow series, mean error ($\beta$) and variability error ($\alpha$) are computed, and then median error values are compared with the original deterministic flow error metrics. Figure 8 shows the improvement of bias ($\alpha$) and variability error ($\beta$) regardless of the performance metric or residual distribution characteristics. Similar to Farmer and Vogel (2016), high flow volume error (percent bias of FHV) and APFB computed with residual incorporated flow series also improve compared to the deterministic flow series from the calibrated models (Figure 9).

The quality of the original deterministic flow simulated by the hydrologic models has little effect on the performance metrics based on the ensemble of residual augmented flows. Since the stochastically generated ensembles do not account for temporal correlation, every ensemble has reduced correlation and deteriorated NSE and KGE metrics. However, the error metric related to the flow duration curve (APFB) is not affected by the lack of correlation because metrics based on FDC do not preserve information regarding the temporal sequence. Although residual augmented flow time-series enhances some of flow metrics, the (temporal) dynamical pattern is not reproduced. These observations point toward the need for careful investigation in interpreting the improvement in model skill especially when different error metrics are considered.

A key issue is the extent to which high flows are represented in the deterministic and the stochastic components. While it is possible to generate ensembles through stochastic simulation of the model residuals (as is done here), and these stochastic simulations improve high-flow error metrics, we will naturally have more confidence in the model simulations if the high flows are well represented in the deterministic model simulations. The use of squared error metrics simply means that a larger part of the high flow signal must be reconstructed via stochastic simulation.

## 6   Conclusions

The use of large sample catchment calibrations of two different hydrologic models with several performance metrics enables us to make robust inferences regarding the effects of the calibration metric on the ability to infer high flow events. Here, we have focused on improving the representation of annual peak flow estimates, as they are important for flood frequency magnitude estimation. We draw the following conclusions from the analysis presented in this paper:

1. The choice of error metric for model calibration impacts high flow estimates very similarly for both models, although mHM provides overall better performance than VIC in terms of all metrics evaluated.

2. Calibration with KGE improves performance as assessed by high flow metrics through improving time-dependent metrics (e.g., variability error score). Adjustment of the scaling factors related to the different KGE components (bias, variability, and correlation terms) can further assist the model simulations in matching certain aspects of flow characteristics. The degree of improvement is, however, model dependent.

3. Application specific metrics can improve estimation of specifically targeted aspects of the system response (here annual peak flows) if used to direct model calibration. However, the use of an application specific metric does not guarantee acceptable performance with regard to other metrics, even those closely related to the application specific metric.

Given that Gupta et al. (2009) shows clear improvement of flow variability estimates by switching the calibration metric from NSE to KGE for a simple rainfall-runoff model similar to the HBV model (Bergström, 1995), and that our results are similar for two relatively more complex models, we can expect that other models would exhibit similar results when using KGE or its scaled variant. If choosing to use an application specific metrics, it seems clear that careful thought needs to be given to the design of the metric if we are to obtain good performance for both the target metric (used for calibration) and other related metrics (used for evaluation). This is important since we wish to increase confidence in the robustness and transferability of the calibrated model- an issue that needs to be examined in more detail.

*Code and data availability.*   Model calibration was performed using MPR-flex available at https://github.com/NCAR/mpr-flex/tree/direct_calib for VIC. mHM is calibrated with the MPR strategy implemented in the mHM http://www.ufz.de/index.php?en=40114. Hydrometeorological data are obtained from a part of Catchment Attributes and Meteorology for Large-sample Studies (CAMELS; Newman et al., 2014; Addor et al., 2017b). Analysis and plotting codes are available at https://github.com/nmizukami/calib4ffa.

*Competing interests.*   The authors declare that they have no conflict of interest.

*Acknowledgements.*   We thank two anonymous referees for their constructive comments and Dr. Ding for his short comment on NSE. The comments helped improve the manuscript, in particular discussion regarding the consideration of deterministic model residuals for error

metric estimates. We also thank Ethan Gutmann and Manabendra Saharia (NCAR) for the earlier discussions on the topic. This work was financially supported by the U.S Army Corps of Engineers Climate Preparedness and Resilience program.

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

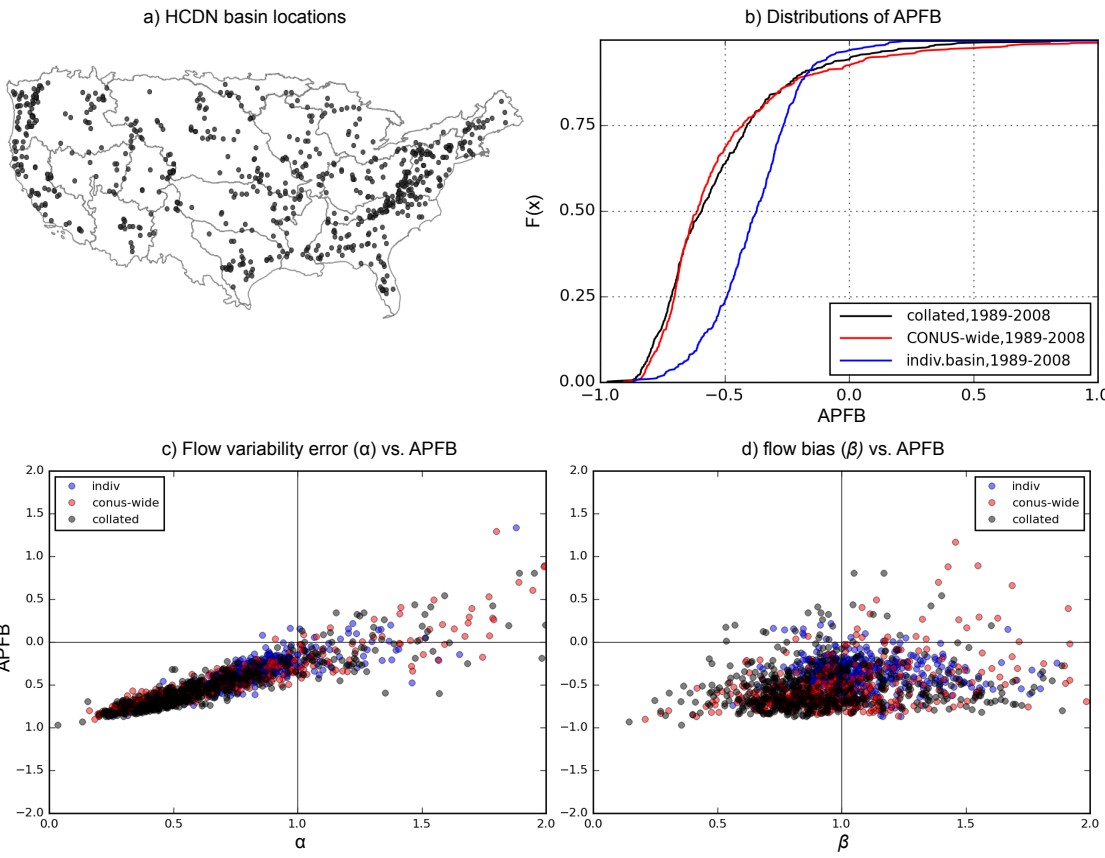

**Figure 1.** Spatial distribution of Hydro-Climate Data Network (HCDN) basins, b) Cumulative distribution of %bias of annual peak flow (APFB) over 1989-2008 simulated with three different sets of VIC parameters used in Mizukami et al. (2017) at HCDN basins. c) Relationships between variability error ($\alpha$: simulation to observation ratio of daily flow variability) with APFB. d) Relationships between mean error ($\beta$: simulation to observation ratio of mean flow) with APFB

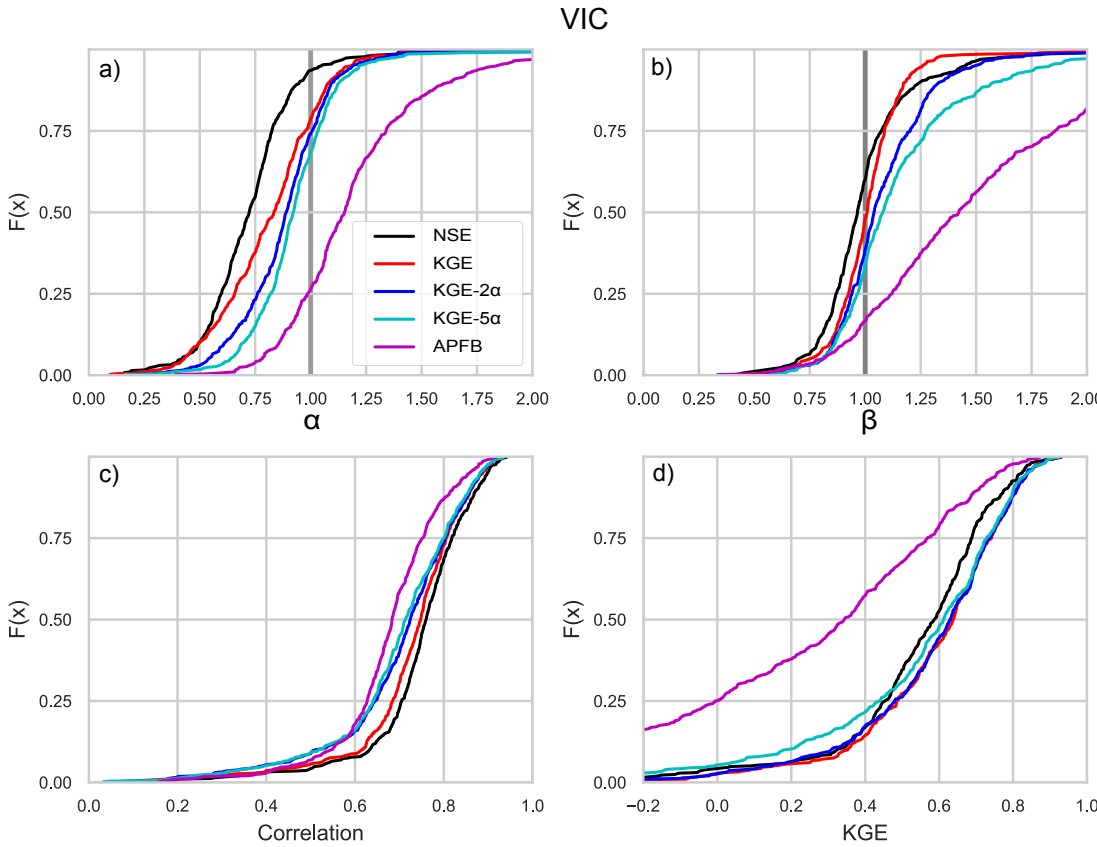

**Figure 2.** Cumulative distributions of a) flow variability errors $\alpha$, b) bias $\beta$, c) linear correlation r, and d) Kling-Gupta Efficiency over the 492 HCDN basin calibrations with 5 performance metrics for evaluation period and VIC.

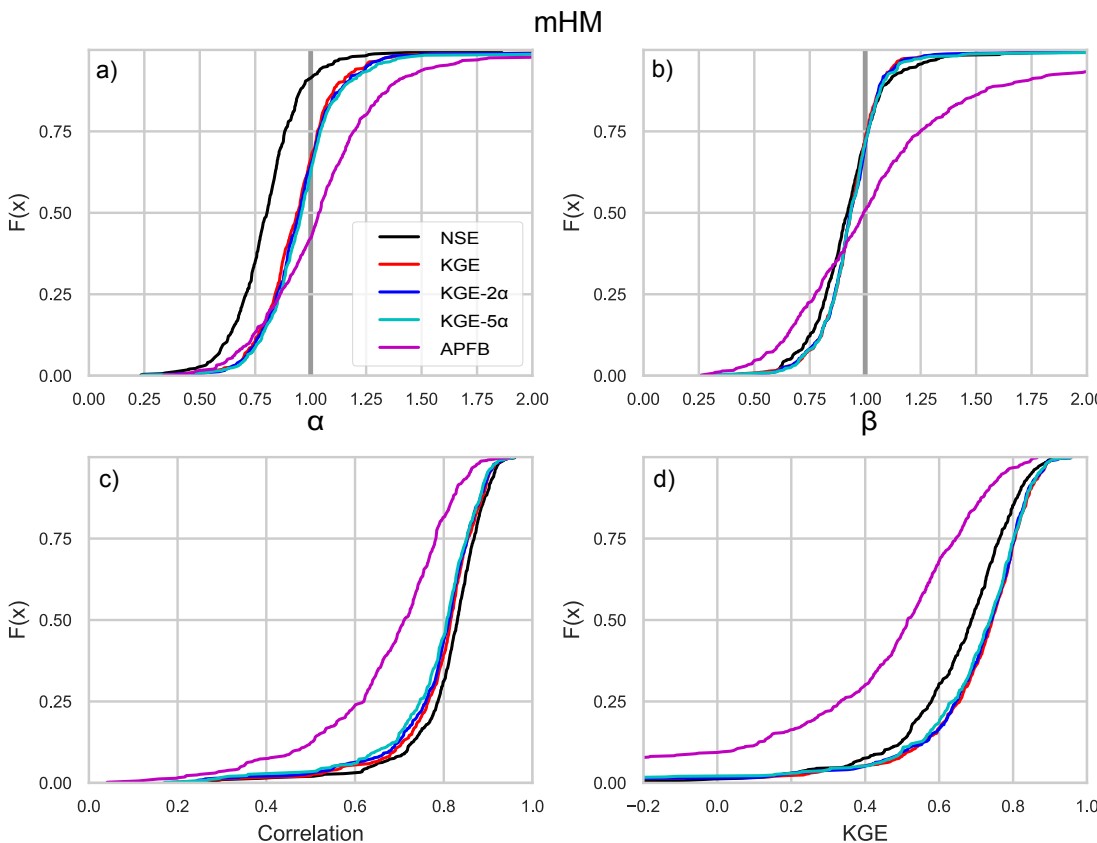

**Figure 3.** The same as Figure 2 except for mHM.

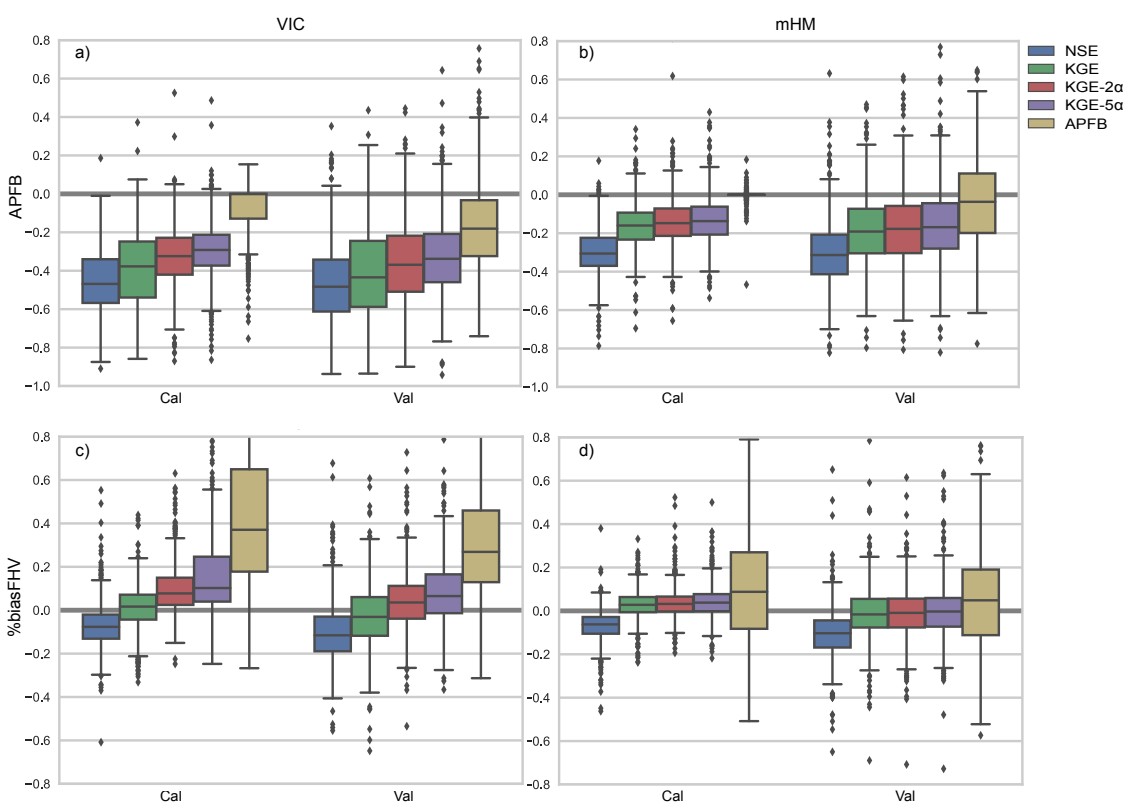

**Figure 4.** Box-plots of %bias of annual peak flow (APFB: top row) and flow volume above 80 percentile flow duration curve (%biasFHV: bottom row) over the 492 HCDN basin calibrations with 5 performance metrics for calibration and evaluation periods and two models. Box width represents inter-quartile range (1st and 3rd quartiles), and lower and upper whiskers are placed at 1.5-time the inter-quartile range.

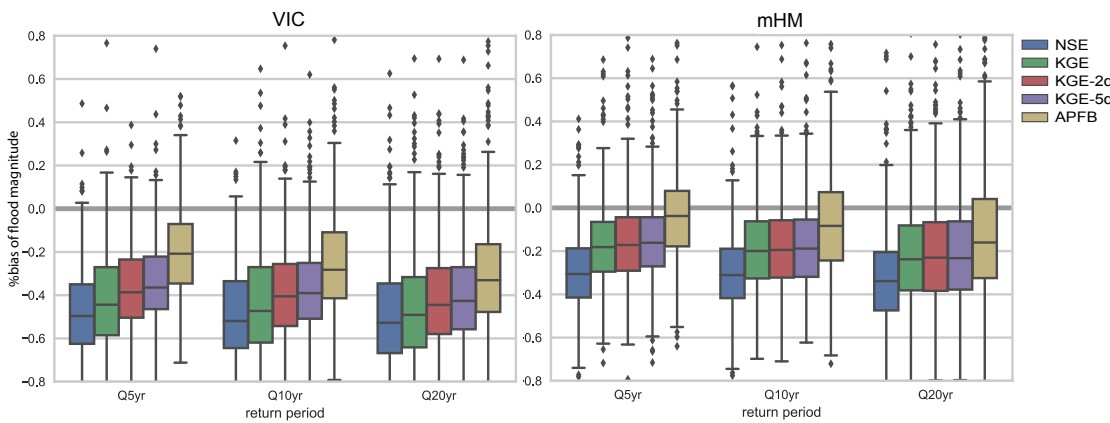

**Figure 5.** Boxplots of percent bias of flood estimates corresponding to three return periods (5-yr, 10-yr and 20-yr) over the 492 HCDN basins for the two models. Boxplot representation is the same as Figure 4

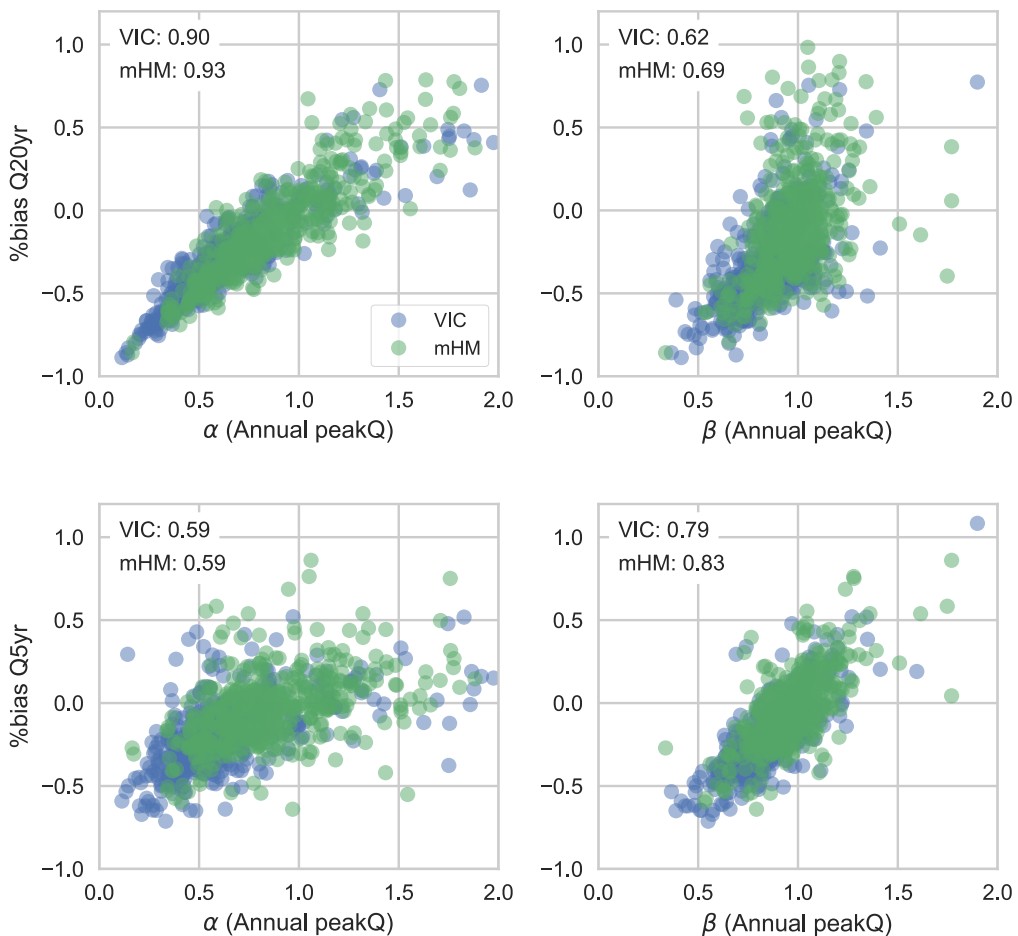

**Figure 6.** Scatter-plots between a) simulation-observation ratio of variability of annual peak flow series ($\alpha$) and %bias of 20-yr flood magnitude, b) simulation-observation ratio of mean annual peak flow series ($\beta$) and %bias of 20-yr flood magnitude, c) and d) are the same as a) and b) except for 5-yr flood magnitudes. Linear correlations between two variables are specified at upper-left corner of each plot.

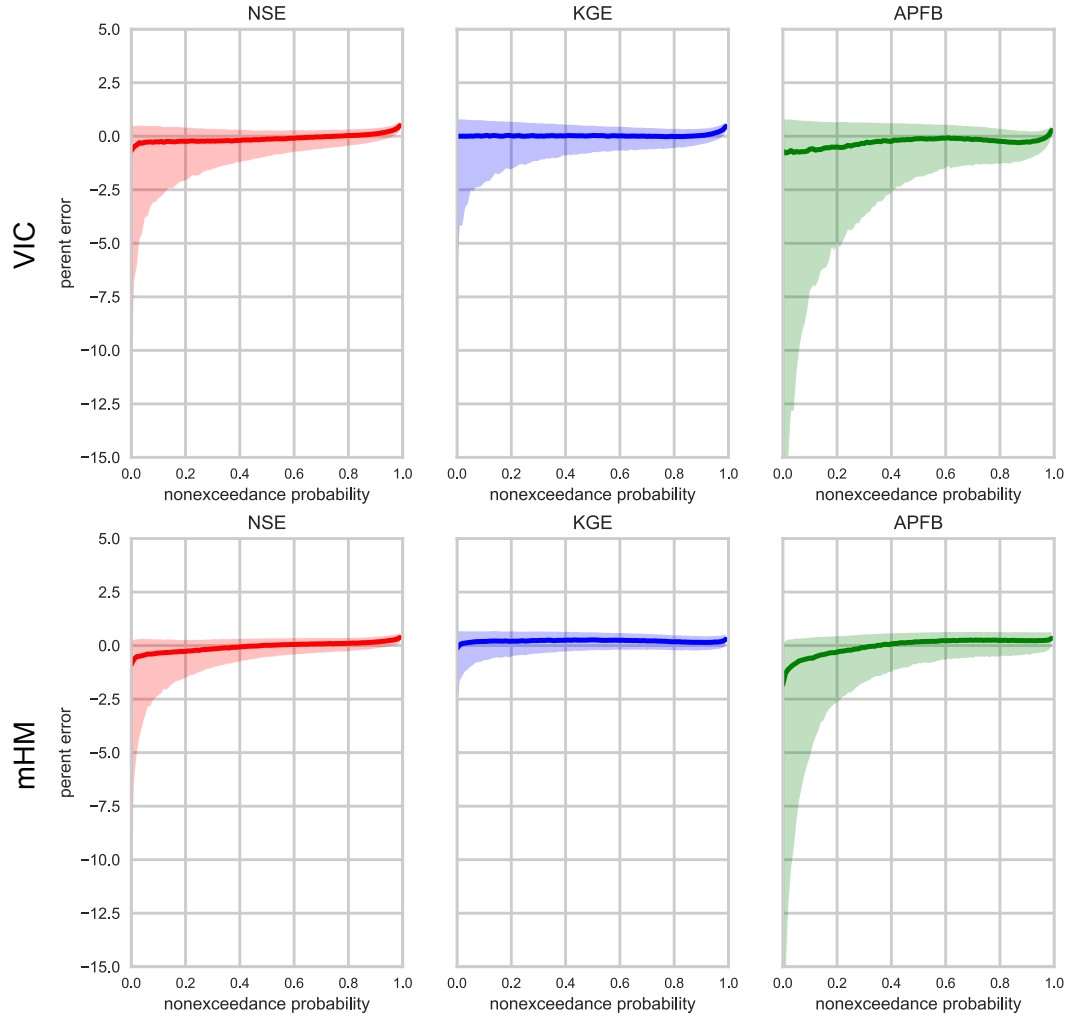

**Figure 7.** Residual distributions conditioned on the non-exceedance probability of the daily flows over the 492 study basins. Analysis are presented for the three calibration performance metrics. Daily residuals are computed based on the observed and simulated flows during the evaluation period.

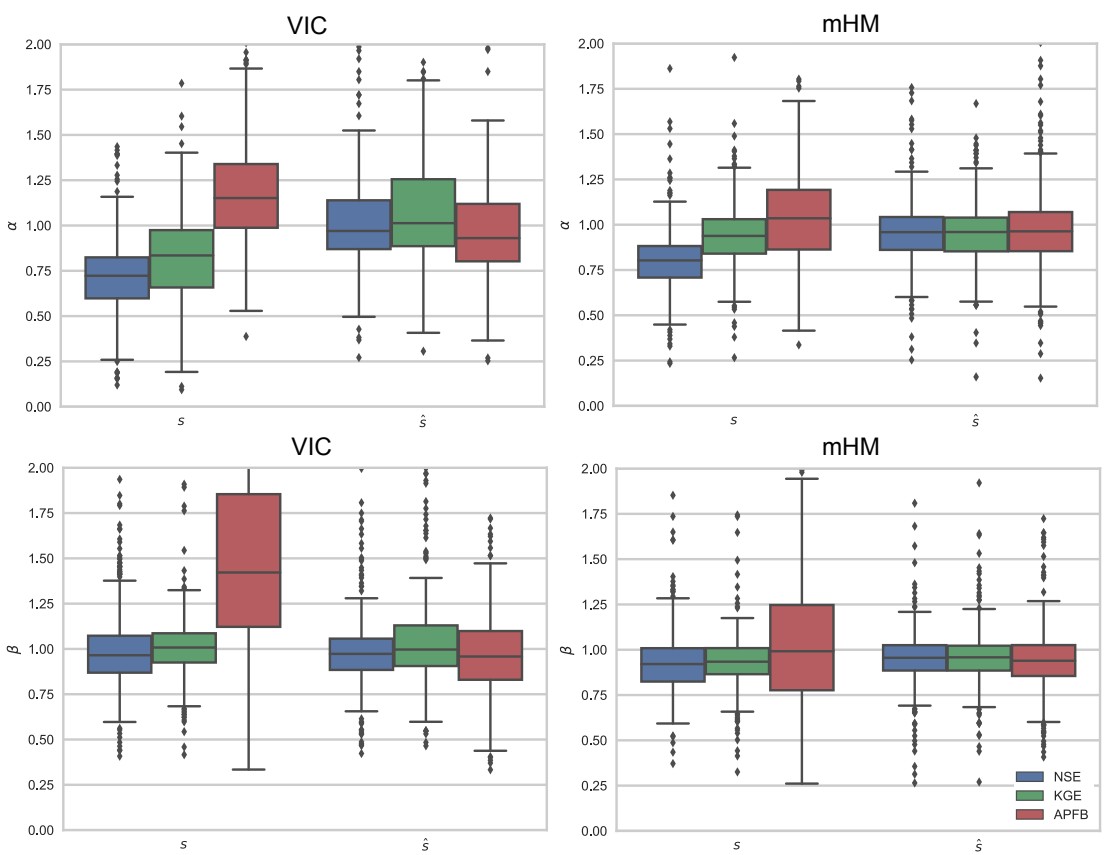

**Figure 8.** Distribution of the two error metrics (top panels: $\alpha$ and bottom panels: $\beta$) computed based on the simulations from NSE, KGE and APFB calibrated models (labeled as s). The distribution of median error metrics (labeled as $\hat{s}$) are based on 100 residual augmented flow series. The evaluation results shown here corresponds to the evaluation period. Box-plot representation is the same as Figure 4.

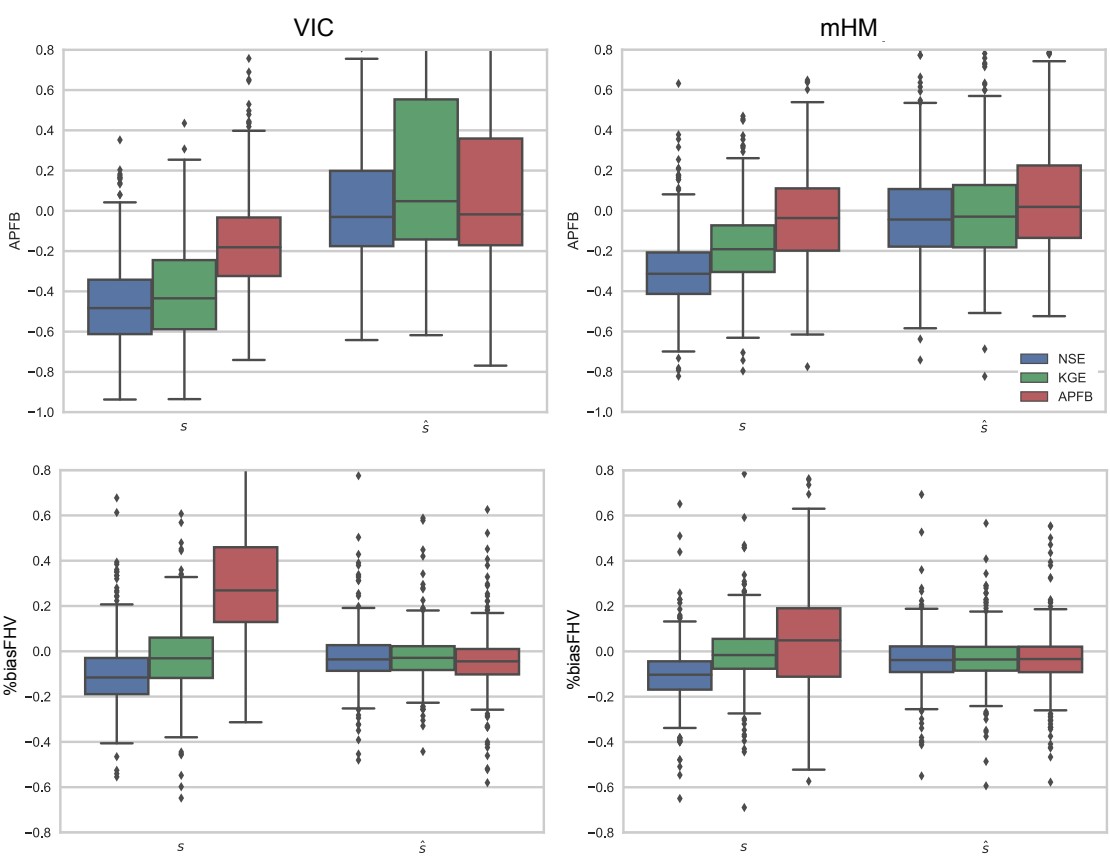

**Figure 9.** The same as Figure 8 except for APFB (top panels) and percent bias in FHV(bottom panels).