# Peer review of "On the choice of calibration metrics for "high flow" estimation using hydrologic models"

_Hydrology and Earth System Sciences, 2018_

## Short Comment (SC1) · 16 Aug 2018

**The NSE criterion**

Having had taken a postgraduate-level course from Nash (1968-69) when he was a visiting professor at my alma mater, the University of Guelph in Ontario, Canada, I believe I was privy to his thinking behind the development of the NSE criterion, a variant of the sum of squared error measure (Nash and Sutcliffe, 1970).

I could be wrong because of the passage of time, but his thinking was that in the absence of a model, the best estimate of the future flows was the average or mean

observed flow value $\overline{O}$, thus its appearance in the denominator of the criterion below (e.g., Ding, 1974, Equation 47):

$$R^2 = 1 - F/F_0, \tag{1}$$

$$F_0 = \sum_{t=0}^{L-1} (Q_{t+1} - \overline{O})^2, \tag{2}$$

$$F = \sum_{t=0}^{L-1} (Q_{t+1} - O_{t+1})^2, \tag{3}$$

in which $R^2$ is the model efficiency; $F_0$ and $F$ are the initial variance from the mean value and the residual variance from the observed values, respectively; $O$ and $Q$ are the observed and simulated flow series, respectively; $t$ is the index of timestep $\Delta t$; and $L$ is the length, or number of ordinates, of the observed hydrograph.

To my way of thinking then, a better estimator would instead be based be a one-step-ahead forecast, $\hat{O}_{t+1} = O_t + (O_t - O_{t-1})$. Replacing the $\overline{O}$ term in Equation (2) by $\hat{O}_{t+1}$ produces Equations (4) and (5):

$$F_1 = \sum_{t=0}^{L-1} (Q_{t+1} - \hat{O}_{t+1})^2, \tag{4}$$

$$R_1^2 = 1 - F/F_1, \tag{5}$$

This would make $F_1$ smaller in value than $F_0$, and the modified NSE criterion, $R_1^2$, a lower score one. The drawback was to always overshoot, by one timestep, the turning points, the peaks and troughs of an observed hydrograph.

**References**

Ding, J. Y.: Variable unit hydrograph, J. Hydrol., 22, 53-69, 1974.

Nash, J. E.: A course of lectures on parametric or analytical hydrology. Great Lakes Institute, University of Toronto, Ont., PR 38, 1968-69.

Nash, J.E. and Sutcliffe, J.V.: River flow forecasting through conceptual models, part I - A discussion of principles, J. Hydrol., 10(3), 282-290, 1970.

---

## Referee Comment (RC1) · Anonymous Referee #1 · 1 Oct 2018

The paper compares the KGE, NSE and a peak flow signature as objective function for the calibration of 2 hydrological models. The paper is well written and clear. However, it does not lead to new results, and the suggestion to abandon NSE in favour of KGE is not well motivated. These points are further elaborated below.

The authors are strongly in favour of KGE vs NSE, as clearly appears from statements such as "Squared error metrics, such as Nash Sutcliffe Efficiency (NSE) and Mean Square Error (MSE), have historically been thought to be useful to reduce simulation errors associated with high flow values (Oudin et al., 2006; Price et al., 2012; Seiller et al., 2017; de Boer-Euser et al., 2017). Although Gupta et al. (2009) showed theoretically how and why the use of NSE and other MSE-based metrics for calibration results in the underestimation of peak flow events, our experience indicates that 20 this notion

continues to persist almost a decade later". One cannot expect NSE to have properties that it is not designed to have, and it would be fair to use such metric in a way that is meaningful and that reflects the theory behind it. The use of sum of squared errors and its rescaled variants is common in statistics, and can be related to precise assumptions about the error. In particular, such objective functions follow the assumption of Normal, zero mean, iid residuals. This is among the simplest assumptions one can make, although often inappropriate, as widely discussed. The properties of a model calibrated using NSE should be considered within the context of this theory. The fact that a deterministic model calibrated using NSE will underestimate the variability of the flow is NOT a design flow of the NSE. It is a characteristic that follows from theory. From theory one can also easily see that it comes to no surprise that the statistics of the deterministic model don't match the statistics of the observed data. They will not match by design. In particular, if the assumption behind squared error metrics is that Qobs=Qmod+eps (with eps N(0,sigma)), it is obvious that the statistics of Qmod are different from the statistics of Qobs. The statistics of Qobs should be compared to the statistics of Qmod+eps. This explains also why, for example, var(Qobs)>var(Qmod). Of course it is, since var(Qobs)=var(Qmod +err)= var(Qmod)+var(err). I can see that the approach of correctly comparing modelled and observed statistics (ie accounting for the error) is almost never followed in the community. This has led to the wrong perception that NS and related metrics somehow don't work.

Therefore, before recommending to switch to other metrics, I would propose the 'old' metrics be tested fairly. Based on this, I have the following suggestions for this paper:

Don't provide poorly grounded indications such as that "squared error type metrics are not suitable for model calibration when the application requires robust high flow performance". NSE and KGE are based on different assumptions, and they should be compared fairly. Even if the KGE results into better performance, one should still note that NSE can be related to properties of the errors, which can be tested and changed if necessary (e.g. one can use the NSE of the sqrt of the flow to reduce

heteroscedasticity).

At present I don't see the novelty of this paper. Most of the statements about the perceived qualities of KGE (part of them debatable, as I explained), are already given in other papers. Conclusion 1 is expected by design of the calibration metrics. Conclusion 2 is unclear. Conclusion 3 is highly debatable as explained.

References:

Farmer, W. H., and R. M. Vogel (2016), On the deterministic and stochastic use of hydrologic models, Water Resour. Res., 52, 5619–5633, doi:10.1002/2016WR019129. Kavetski, D., F. Fenicia, P. Reichert, and C. Albert (2018), Signature-Domain Calibration of Hydrological Models Using Approximate Bayesian Computation: Theory and Comparison to Existing Applications, Water Resour Res, 54(6), 4059-4083.

---

## Referee Comment (RC2) · Anonymous Referee #2 · 4 Nov 2018

**1 OVERALL RECOMMENDATION**

The manuscript addresses the important topic of the choice of calibration metrics (CM) to be used for rainfall-runoff modeling, and presents results obtained on 492 US catchments. I found the paper interesting, including relevant references. If the presented results are not highly original, the paper is, in his present form, an excellent illustration of the limitation of the use of Nash and Sutcliffe efficiency metric (NSE, 1970) for model calibration. Nevertheless, I do have major comments on the used dataset, the applied methodology and the discussion part. Thus, I recommend to accept the manuscript in HESS with major revisions detailed below.

**2. GENERAL COMMENTS**

**2.1 Description of the studied catchments**

Even though the objective of such "large-sample hydrology paper" is not to present results obtained on a limited number of catchments having the same hydro-meteorological characteristics but to have general conclusions on rainfall-runoff modeling, I think the diversity of the studied catchments has to be addressed and quantified. This description is lacking right now in the paper. A presentation of the general characteristics of the studied catchments should be added in the paper, in order to understand the variability of catchments characteristics (catchment area, runoff coefficient, mean annual solid precipitation, etc.), especially in the context of flood modeling: what are different flood processes and dynamics included within this catchments sets (flash floods, snowmelt floods, rain-on-snow floods, groundwater floods, etc.)? Moreover, the timestep considered in the two rainfall-runoff models is not stated in the paper and should be mentioned. Are the models working at daily timestep? Is this timestep consistent with the flood dynamics of every studied catchment?

**2.2 Split-sample test**

For every catchment, the calibration and validation periods are the same time-periods, 1999-2008 and 1989-1999, respectively. I think that performing a basic split and sample test (Klemeš, 1986) on each catchment would be particularly interesting in this context, especially to address temporal (in)stability of parameter sets obtained with particular CM (topic partially addressed page 6, line 12).

**2.3 List of the studied CM**

The paragraph listing the studied CM (page 5, lines 5 to 16) is unclear and would be easier to understand if a list (or table) of the five studied CM was added.

**2.4 "Application-specific" or "hydrologic signature"?**

From page 5 to the end of the paper, APFB is named as an "application-specific" metric, while being introduced as an "hydrologic signature" (see definition of "hydrologic

signature" in the paper introduction, page 2 lines 10 to 24) in the paper objective presentation. What is the difference between an "application specific" and a "hydrologic signature" CM in this context? Finally, is APFB an "application specific" or an "hydrologic signature" CM? Could you address this point?

**2.5 Impact of the KGE scaling factors**

The limited impact of the different KGE scaling factors used in the paper is very little discussed, while being particularly interesting. This point has to be discussed in the paper. Moreover, why not trying another combination with a larger variability ratio scaling factor, such as (Sr=1, Saplha=20, Sbeta=1), to assess a potential significant improvement of annual peak flow bias? What about another test with (Sr=0.1, Saplha=1, Sbeta=0.1) or even (Sr=0, Saplha=1, Sbeta=0)?

**2.6 Figures**

In general, the presentation of the figures could be improved for a better understanding:

- The performance metrics plotted have different names in the axis labels and in the figure legends (e.g. it is not explicit that "%biasFHV" is equal to "percentage bias of flow volume above 80 percentile flow duration curve" in the Figure 4) ;

- The name/typography of several performance metrics is changing over figures ("%bias Qpeak" on Figure 2 but "%biasQpeak" on Figure 4) ;

- Why not using Greek letters in Figure 2 x-axis?

- The link between the five CM and the figure legend is never clearly stated, and for example, the reader has to guess that "kge_2alpha" is equal to (Sr=1, Saplha=2, Sbeta=1).

- How boxplots have been constructed? What are the outlier points plotted below and over the boxplots?

**3 SPECIFIC COMMENTS**

1. Page 4, line 27: please change (Maurel et al., 2002) into Maurel et al. (2002).

2. Page 13, figure 2: please state in the figure legend that results presented in this figure are obtained with the VIC model.

**4 REFERENCES**

Klemeš, V., 1986. Operational testing of hydrological simulation models. Hydrological Sciences Journal 31, 13. https://doi.org/10.1080/02626668609491024.

Nash, J.E., Sutcliffe, J.V., 1970. River flow forecasting through conceptual models part I – A discussion of principles. Journal of Hydrology 10, 282–290. https://doi.org/10.1016/0022-1694(70)90255-6.

---

## Author Comment (AC1) · 23 Nov 2018

**Response to Dr. John Ding**

Here we provide our response to comment from Dr. Ding including the plan to revise the manuscript in response to his comments. The original review comments are in black, and our responses in red.

SC1 The NSE criterion

Having had taken a postgraduate-level course from Nash (1968-69) when he was a visiting professor at my alma mater, the University of Guelph in Ontario, Canada, I believe I was privy to his thinking behind the development of the NSE criterion, a variant of the sum of squared error measure (Nash and Sutcliffe, 1970).

I could be wrong because of the passage of time, but his thinking was that in the absence of a model, the best estimate of the future flows was the average or mean observed flow value $\bar{O}$, thus its appearance in the denominator of the criterion below (e.g., Ding, 1974, Equation 47):

$$R^2 = 1 - F/F_0$$

$$F_0 = \sum_{t=0}^{L-1}(Q_{t+1} - \bar{O})^2$$

$$F = \sum_{t=0}^{L-1}(Q_{t+1} - O_{t+1})^2$$

in which $R^2$ is the model efficiency; $F_0$ and F are the initial variance from the mean value and the residual variance from the observed values, respectively; $O$ and $Q$ are the observed and simulated flow series, respectively; $t$ is the index of timestep $\Delta t$; and $L$ is the length, or number of ordinates, of the observed hydrograph.

To my way of thinking then, a better estimator would instead be based be a one-step ahead forecast $\hat{O}_{t+1} = O_t + (O_t - O_{t-1})$. Replacing the $O$ term in Equation (2) by $\hat{O}_{t+1}$ produces Equations (4) and (5):

$$F_1 = \sum_{t=0}^{L-1}\left(Q_{t+1} - \hat{O}_{t+1}\right)^2$$

$$R_1^2 = 1 - F/F_1$$

This would make $F_1$ smaller in value than $F_0$, and the modified NSE criterion, $R_1^2$, a lower score one. The drawback was to always overshoot, by one timestep, the turning points, the peaks and troughs of an observed hydrograph.

References

Ding, J. Y.: Variable unit hydrograph, J. Hydrol., 22, 53-69, 1974.

Nash, J. E.: A course of lectures on parametric or analytical hydrology. Great Lakes Institute, University of Toronto, Ont., PR 38, 1968-69.
Nash, J.E. and Sutcliffe, J.V.: River flow forecasting through conceptual models, part I

- A discussion of principles, J. Hydrol., 10(3), 282-290, 1970.

First of all, we appreciate your efforts to share the history of earlier work on optimization metric development. Based on the Nash and Sutcliffe's paper, development of NSE was motivated for the model optimization and the NSE efficiency given for a particular model is a ratio of the model skill, here sum of squared of error, to the simplest flow estimation (i.e., observed mean flow).

More generally the NSE is formulated as a skill score $S=1-e/e_{ref}$, where e is some error metric and $e_{ref}$ is the reference benchmark. While the observed variance is used as a benchmark in the NSE, other benchmarks can be used as well (e.g., see Schaefli and Gupta, 2007). To our understanding, your proposal is to use persistence as a benchmark, an approach used in the climate community.

NSE has been used predominantly for model optimization (esp., hydrologic models) for their practical applications. In the meantime, there has been substantial studies on the metrics along with evolution of optimization techniques (i.e., multi-criteria-based optimization). This has been discussed in Gupta et al., 2009.

If your intent is to advocate for a wider set of benchmarks, then we wholeheartedly agree on this point (e.g., Pappenberger et al., 2015); however, this is not the main objective of this contribution. We consider adding some brief descriptions on the meaning of NSE along with the historical developments of the optimization metrics.

**References**

Gupta, H. V., Kling, H., Yilmaz, K. K., and Martinez, G. F., 2009. Decomposition of the mean squared error and NSE performance criteria: Implications for improving hydrological modelling, *Journal of Hydrology*, 377, pp.80–91.

Pappenberger, F., Ramos, M.H., Cloke, H.L., Wetterhall, F., Alfieri, L., Bogner, K., Mueller, A. and Salamon, P., 2015. How do I know if my forecasts are better? Using benchmarks in hydrological ensemble prediction. *Journal of Hydrology*, *522*, pp.697-713.

Schaefli, B. and Gupta, H.V., 2007. Do Nash values have value?. *Hydrological Processes: An International Journal*, *21*(15), pp.2075-2080.

---

## Author Comment (AC3) · 23 Nov 2018

The comment was uploaded in the form of a supplement:
https://www.hydrol-earth-syst-sci-discuss.net/hess-2018-391/hess-2018-391-AC3-supplement.pdf

---

## Author Response (AR1)

**Details of revision based on comments from referees**

This documents how we addressed the comments in the revised manuscript. The original review comments are in black, and our responses in red. We also made additional textual revision throughout the text.

**RC1**

The paper compares the KGE, NSE and a peak flow signature as objective function for the calibration of 2 hydrological models. The paper is well written and clear. However, it does not lead to new results, and the suggestion to abandon NSE in favour of KGE is not well motivated. These points are further elaborated below.

The authors are strongly in favour of KGE vs NSE, as clearly appears from statements such as "Squared error metrics, such as Nash Sutcliffe Efficiency (NSE) and Mean Square Error (MSE), have historically been thought to be useful to reduce simulation errors associated with high flow values (Oudin et al., 2006; Price et al., 2012; Seiller et al., 2017; de Boer-Euser et al., 2017). Although Gupta et al. (2009) showed theoretically how and why the use of NSE and other MSE-based metrics for calibration results in the underestimation of peak flow events, our experience indicates that this notion continues to persist almost a decade later".

One cannot expect NSE to have properties that it is not designed to have, and it would be fair to use such metric in a way that is meaningful and that reflects the theory behind it.

The use of sum of squared errors and its rescaled variants is common in statistics, and can be related to precise assumptions about the error. In particular, such objective functions follow the assumption of Normal, zero mean, iid residuals. This is among the simplest assumptions one can make, although often inappropriate, as widely discussed. The properties of a model calibrated using NSE should be considered within the context of this theory. The fact that a deterministic model calibrated using NSE will underestimate the variability of the flow is NOT a design flaw of the NSE. It is a characteristic that follows from theory. From theory one can also easily see that it comes to no surprise that the statistics of the deterministic model don't match the statistics of the observed data. They will not match by design. In particular, if the assumption behind squared error metrics is that Qobs=Qmod+eps (with eps N(0,sigma)), it is obvious that the statistics of Qmod are different from the statistics of Qobs. The statistics of Qobs should be compared to the statistics of Qmod+eps. This explains also why, for example, var(Qobs)>var(Qmod). Of course it is, since var(Qobs)=var(Qmod +err)= var(Qmod)+var(err). I can see that the approach of correctly comparing modelled and observed statistics (ie accounting for the error) is almost never followed in the community. This has led to the wrong perception that NS and related metrics somehow don't work.

Therefore, before recommending to switch to other metrics, I would propose the 'old' metrics be tested fairly. Based on this, I have the following suggestions for this paper: Don't provide poorly grounded indications such as that "squared error type metrics are not suitable for model calibration when the application requires robust high flow performance". NSE and KGE are based on different assumptions, and they should be compared fairly. Even if the KGE results into better performance, one should still note that NSE can be related to properties of the errors, which can be tested and changed if necessary (e.g. one can use the NSE of the sqrt of the flow to reduce heteroscedasticity).

At present I don't see the novelty of this paper. Most of the statements about the perceived qualities of KGE (part of them debatable, as I explained), are already given in other papers. Conclusion 1 is expected by design of the calibration metrics. Conclusion 2 is unclear. Conclusion 3 is highly debatable as explained.

**References:**

Farmer, W. H., and R. M. Vogel (2016), On the deterministic and stochastic use of hydrologic models, Water Resour. Res., 52, 5619–5633, doi:10.1002/2016WR019129.

Kavetski, D., F. Fenicia, P. Reichert, and C. Albert (2018), Signature-Domain Calibration of Hydrological Models Using Approximate Bayesian Computation: Theory and Comparison to Existing Applications, Water Resour Res, 54(6), 4059-4083.

We very much appreciate comments coming from different perspective related to modeled flow metrics. We completely agree with the main comment – that sum-of-squared error metrics commonly used in optimization reduce the variance by design, and representing the flow statistical moments and extremes requires stochastically simulating the error term. We carefully reviewed the paper by Famer and Vogel (2016) who discuss about the stochastic estimation of model errors and effect of residuals on high flow metrics. The ideas illustrated in the Farmer and Vogel (2016) paper help us better frame our contribution.

In the revised discussion section (section 5.2 P8-9), we analyze the distributions of errors for models calibrated using three objective functions (KGE/NSE/APFB), as well as examining the flow time series metrics (mean and variability) and high flow metrics for deterministic KGE/NSE/APFB calibrated flow and ensemble flows based on stochastically generated error added to respective calibrated streamflow simulations. As you see in the discussion, we observe that the ensemble of residual reintroduced flows does improve the metrics (mean, variances, and high flow statistics) regardless of residual distribution of the deterministic flow. However, the dynamical property (i.e., temporal pattern) deteriorates due to lack of temporal correlation in the synthetic flow sequences. The method for stochastic residual generation uses random sampling based on flow magnitudes and requires incorporation of auto correlation properties.

While we can arrive at similar conclusions to Farmer and Vogel (2016), we feel that obtaining improved deterministic flow simulation through model calibration is important because of improvement of application specific flow metrics through improving the magnitude, variability, temporal correlation. And we feel that impacts of performance metric choice on deterministic flow metrics are still not well appreciated by the broader community. We hope that our paper provides additional explanations of unintended consequences of model calibration decisions.

We have slightly revised the conclusions in the revised manuscript. Our main point however still remains – alternatives to sum-of-squared error metrics can improve the deterministic component of the model simulations, especially for high flows. This is important since most hydrologic modeling applications only consider the deterministic component.

**RC2**

**1 OVERALL RECOMMENDATION**

The manuscript addresses the important topic of the choice of calibration metrics (CM) to be used for rainfall-runoff modeling, and presents results obtained on 492 US catchments. I found the paper interesting, including relevant references. If the presented results are not highly original, the paper is, in his present form, an excellent illustration of the limitation of the use of Nash and Sutcliffe efficiency metric (NSE, 1970) for model calibration. Nevertheless, I do have major comments on the used dataset, the applied methodology and the discussion part. Thus, I recommend to accept the manuscript in HESS with major revisions detailed below.

**2. GENERAL COMMENTS**

**2.1 Description of the studied catchments**

Even though the objective of such "large-sample hydrology paper" is not to present results obtained on a limited number of catchments having the same hydrometeorological characteristics but to have general conclusions on rainfall-runoff modeling, I think the diversity of the studied catchments has to be addressed and quantified.

This description is lacking right now in the paper. A presentation of the general characteristics of the studied catchments should be added in the paper, in order to understand the variability of catchments characteristics (catchment area, runoff coefficient, mean annual solid precipitation, etc.), especially in the context of flood modeling: what are different flood processes and dynamics included within this catchments sets (flash floods, snowmelt floods, rain-on-snow floods, groundwater floods, etc.)? Moreover, the timestep considered in the two rainfall-runoff models is not stated in the paper and should be mentioned. Are the models working at daily timestep? Is this timestep consistent with the flood dynamics of every studied catchment?

Yes, one of main objectives of large sample basin study is to generalize the conclusions drawn regarding hydrologic modeling evaluations (Gupta et al 2014). This manuscript used a subset (492 out of 671) the catchments presented by Addor et al., (2018), who describe in detail the variability of climate/geophysical/hydrologic characteristics for the 671 US catchments. Our basin selections are also spread over the CONUS; therefore, distributions of basin characteristics are similar to Addor et al., (2018). We decided to avoid repetitive summaries and figures.

We looked at spatial pattern of the model skills in addition to the distributions (See Figs R1 and R2). There is little distinct spatial pattern in the APFB (%bias Qpeak in Fig R2). This indicates that catchment characteristics have less effects than the performance metrics P6, L25. We mentioned this text in L-235-236 (not shown in Figures in the revised manuscript).

As for the time step, we performed daily simulation for calibration in this study as we stated in P5, L10. Though calibrated parameter values may not be consistent for different temporal resolutions, the trend in calibration performances across the different performance metrics should be preserved regardless of the time steps. Moreover, the theory of algebraic decomposition of NSE is independent of the time step.

%bias of annual peak Q, NSE calibration

---

## Author Response (AR2)

**Response to referee comments**

Here we provide our response to one comment on the catchment variability in terms of flood mechanisms from referee 2. The original review comment is shown in black, and our response is in red.

**RC2**

The authors adequately addressed my previous comments. I still have a minor comment about the variability of catchments characteristics in the context of flood modeling: what are the different flood processes and dynamics included within this catchment set (flash floods, snowmelt floods, rain-on-snow floods, groundwater floods, etc.)? Is daily timestep consistent with the flood dynamics of every studied catchments?

We appreciate referee 2 for careful reading our previous revision. Large flood mechanisms have been discussed in the context of US large sample hydrology (Berghuijs et al. 2016). The most important factors to the flood mechanism, beside extreme precipitation event and rapid snowmelt, are the degree of soil moisture saturation (Berghuijs et al. 2016; Slater and Wilby 2017; Tarasova et al. 2018). Antecedent soil moisture state also plays an important role for flash flood events- a flood that occurs within minutes or a few hours of extreme rainfall in a short period of time, generally less than 6 hours. Simulation of flash flood does require sub-daily (e.g., hourly) hydrologic model runs.  Also, hourly simulations produce higher values for high return period than daily simulation using the same parameters (e.g., Zhu et al. 2018).  If we repeat the analysis performed in the paper with hourly simulations (including the model calibrations), however, they produce the same trends (impact of calibration metrics on high flow simulations) as daily simulations, though accuracy may differ. Furthermore, recent studies (Ding et al. 2016; Dieter and Arns 2003) showed that instantaneous peak flow (at sub-daily scale) is predicted well using daily mean flow, justifying daily model simulations still provide useful information for instantaneous peak flow estimates.

To incorporate this comment into further revision, we added sentences that describe above discussion in Line 185-192 and Line 194-196.

Berghuijs, W. R., R. A. Woods, C. J. Hutton, and M. Sivapalan, 2016: Dominant flood generating mechanisms across the United States. *Geophys. Res. Lett.*, **43**, 4382–4390, doi:10.1002/2016GL068070. https://doi.org/10.1002/2016GL068070.

Dieter, F. H., and S. A. Arns, 2003: Estimating Instantaneous Peak Flow from Mean Daily Flow Data. *J. Hydrol. Eng.*, **8**, 365–369, doi:10.1061/(ASCE)1084-0699(2003)8:6(365). https://doi.org/10.1061/(ASCE)1084-0699(2003)8:6(365).

Ding, J., M. Wallner, H. Müller, and U. Haberlandt, 2016: Estimation of instantaneous peak flows from maximum mean daily flows using the HBV hydrological model. *Hydrol. Process.*, **30**, 1431–1448, doi:10.1002/hyp.10725. https://doi.org/10.1002/hyp.10725.

Slater, L. J., and R. L. Wilby, 2017: Measuring the changing pulse of rivers. *Science (80-. ).*, **357**, 552 LP – 552, doi:10.1126/science.aao2441. http://science.sciencemag.org/content/357/6351/552.abstract.

Tarasova, L., S. Basso, C. Poncelet, and R. Merz, 2018: Exploring Controls on Rainfall-Runoff Events: 2. Regional Patterns and Spatial Controls of Event Characteristics in Germany. *Water Resour. Res.*, **54**, 7688–7710, doi:10.1029/2018WR022588. https://doi.org/10.1029/2018WR022588.

Zhu, Z., D. B. Wright, and G. Yu, 2018: The Impact of Rainfall Space-Time Structure in Flood Frequency Analysis. *Water Resour. Res.*, **54**, 8983–8998, doi:10.1029/2018WR023550. https://doi.org/10.1029/2018WR023550.

[revised manuscript text omitted]

$$\alpha = \sigma_s/\sigma_o, \quad \beta = \mu_s/\mu_o$$

Eq. (1)

where $S_r$, $S_\alpha$ and $S_\beta$ are user specified scaling factors for the correlation ($r$), variability ratio ($\alpha$), and mean ratio ($\beta$) terms; $\sigma_s$ and $\sigma_o$ are the standard deviation values for the simulated and observed responses respectively, and $\mu_s$ and $\mu_o$ are the corresponding mean values. In a balanced formulation, $S_r$, $S_\alpha$ and $S_\beta$ are all set to 1.0. By changing the relative sizes of the $S_r$, $S_\alpha$ or $S_\beta$ weights, the calibration can be altered to more strongly emphasize the reproduction of flow timing, statistical variability, or long-term water balance.

[12]  The results of the *Mizukami et al. (2017)* large sample study motivated us to carry out further experiments to investigate how the choice of performance metric affects the estimation of peak and high flow. Here, we examine the extent to which altering the scale factors in KGE can result in improved high flow simulations compared to NSE. We also examine the results provided by use of an application specific metric, here taken as the %bias in annual peak flows.

**3    Models, Datasets and Methods**

[13]  We use two hydrologic models; VIC and mHM. The VIC model, which includes explicit soil-vegetation-snow processes, has been used for a wide range of hydrologic applications, and has recently been evaluated in large-sample predictability benchmark study (*Newman et al. 2017*). The mHM model has been shown to provide robust hydrologic simulations over both Europe and the US (*Kumar et al. 2013a; Rakovec et al. 2016a*) and is currently being used in application studies *(e.g., Samaniego et al. 2018; Thober et al. 2018)*. We use daily observed streamflow data at the HCDN basins and daily basin meteorological data from *Maurer et al. (2002)* for the period from 1980 through 2008, as compiled by the CONUS large sample basin dataset over a wide range of climate regimes (*Addor et al. 2017; Newman et al. 2014*). Interested readers may refer to *Addor et al. 2017 and Newman et al. 2014* for more details and insights into different physiographic and hydro-climatic chcearacteristics of the study basins. 
[revised manuscript text omitted]

de Boer-Euser, T., and Coauthors, 2017: Looking beyond general metrics for model comparison -- lessons from an international model intercomparison study. *Hydrol. Earth Syst. Sci.*, **21**, 423–440, doi:10.5194/hess-21-423-2017. https://www.hydrol-earth-syst-sci.net/21/423/2017/.

Bourgin, F., V. Andréassian, C. Perrin, and L. Oudin, 2015: Transferring global uncertainty estimates from gauged to ungauged catchments. *Hydrol. Earth Syst. Sci.*, **19**, 2535–2546, doi:10.5194/hess-19-2535-2015. https://www.hydrol-earth-syst-sci.net/19/2535/2015/.

Cuntz, M., J. Mai, L. Samaniego, M. Clark, V. Wulfmeyer, O. Branch, S. Attinger, and S. Thober, 2016: The impact of standard and hard-coded parameters on the hydrologic fluxes in the Noah-MP land surface model. *J. Geophys. Res.*, **121**, 10,676-10,700, doi:10.1002/2016JD025097.

Dieter, F. H., and S. A. Arns, 2003: Estimating Instantaneous Peak Flow from Mean Daily Flow Data. *J. Hydrol. Eng.*, **8**, 365–369, doi:10.1061/(ASCE)1084-0699(2003)8:6(365). https://doi.org/10.1061/(ASCE)1084-0699(2003)8:6(365).

Ding, J., M. Wallner, H. Müller, and U. Haberlandt, 2016: Estimation of instantaneous peak flows from maximum mean daily flows using the HBV hydrological model. *Hydrol. Process.*, **30**, 1431–1448, doi:10.1002/hyp.10725. https://doi.org/10.1002/hyp.10725.

Elsner, M., and Coauthors, 2010: Implications of 21st century climate change for the hydrology of Washington State. *Clim. Change*, **102**, 225–260, doi:10.1007/s10584-010-9855-0. http://dx.doi.org/10.1007/s10584-010-9855-0.

Elsner, M. M., S. Gangopadhyay, T. Pruitt, L. D. Brekke, N. Mizukami, and M. P. Clark, 2014: How Does the Choice of Distributed Meteorological Data Affect Hydrologic Model Calibration and Streamflow Simulations? *J. Hydrometeorol.*, **15**, 1384–1403, doi:10.1175/jhm-d-13-083.1. http://dx.doi.org/10.1175/JHM-D-13-083.1.

Farmer, W. H., and R. M. Vogel, 2016: On the deterministic and stochastic use of hydrologic models. *Water Resour. Res.*, **52**, 5619–5633, doi:10.1002/2016WR019129. https://doi.org/10.1002/2016WR019129.

Garcia, F., N. Folton, and L. Oudin, 2017: Which objective function to calibrate rainfall–runoff models for

low-flow index simulations? *Hydrol. Sci. J.*, **62**, 1149–1166, doi:10.1080/02626667.2017.1308511.

Gupta, H. V., S. Sorooshian, and P. O. Yapo, 1998: Toward improved calibration of hydrologic models: Multiple and noncommensurable measures of information. *Water Resour. Res.*, **34**, 751–763, doi:10.1029/97wr03495. http://dx.doi.org/10.1029/97WR03495.

Gupta, H. V., K. J. Beven, and T. Wagener, 2006: Model Calibration and Uncertainty Estimation. *Encycl. Hydrol. Sci.*, doi:doi:10.1002/0470848944.hsa138. https://doi.org/10.1002/0470848944.hsa138.

Gupta, H. V, T. Wagener, and Y. Liu, 2008: Reconciling theory with observations: elements of a diagnostic approach to model evaluation. *Hydrol. Process.*, **22**, 3802–3813, doi:10.1002/hyp.6989. http://dx.doi.org/10.1002/hyp.6989.

——, H. Kling, K. K. Yilmaz, and G. F. Martinez, 2009: Decomposition of the mean squared error and NSE performance criteria: Implications for improving hydrological modelling. *J. Hydrol.*, **377**, 80–91, doi:http://dx.doi.org/10.1016/j.jhydrol.2009.08.003. http://www.sciencedirect.com/science/article/pii/S0022169409004843.

Gupta, H. V, C. Perrin, G. Blöschl, A. Montanari, R. Kumar, M. Clark, and V. Andréassian, 2014: Large-sample hydrology: a need to balance depth with breadth. *Hydrol. Earth Syst. Sci.*, **18**, 463–477, doi:10.5194/hess-18-463-2014. http://www.hydrol-earth-syst-sci.net/18/463/2014/.

Kavetski, D., F. Fenicia, P. Reichert, and C. Albert, 2018: Signature-Domain Calibration of Hydrological Models Using Approximate Bayesian Computation: Theory and Comparison to Existing Applications. *Water Resour. Res.*, **54**, 4059–4083, doi:10.1002/2017WR020528. https://doi.org/10.1002/2017WR020528.

Klemes, V., 1986: Operational testing of hydrological simulation models. *Hydrol. Sci. J.*, **31**, 13–24, doi:10.1080/02626668609491024. https://doi.org/10.1080/02626668609491024.

Kumar, R., L. Samaniego, and S. Attinger, 2010: The effects of spatial discretization and model parameterization on the prediction of extreme runoff characteristics. *J. Hydrol.*, **392**, 54–69, doi:http://dx.doi.org/10.1016/j.jhydrol.2010.07.047. http://www.sciencedirect.com/science/article/pii/S0022169410004865.

Kumar, R., B. Livneh, and L. Samaniego, 2013a: Toward computationally efficient large-scale hydrologic predictions with a multiscale regionalization scheme. *Water Resour. Res.*, **49**, 5700–5714, doi:10.1002/wrcr.20431. http://dx.doi.org/10.1002/wrcr.20431.

Kumar, R., L. Samaniego, and S. Attinger, 2013b: Implications of distributed hydrologic model parameterization on water fluxes at multiple scales and locations. *Water Resour. Res.*, **49**, 360–379, doi:10.1029/2012wr012195. http://dx.doi.org/10.1029/2012WR012195.

Liang, X., D. P. Lettenmaier, E. F. Wood, and S. J. Burges, 1994: A simple hydrologically based model of land surface water and energy fluxes for general circulation models. *J. Geophys. Res.*, **99**, 14415–14428, doi:10.1029/94jd00483. http://dx.doi.org/10.1029/94JD00483.

Maurer, E. P., A. W. Wood, J. C. Adam, D. P. Lettenmaier, and B. Nijssen, 2002: A Long-Term Hydrologically Based Dataset of Land Surface Fluxes and States for the Conterminous United

States. *J. Clim.*, **15**, 3237–3251, doi:10.1175/1520-0442(2002)015<3237:althbd>2.0.co;2. http://dx.doi.org/10.1175/1520-0442(2002)015%3C3237:ALTHBD%3E2.0.CO;2.

Mendoza, P. A., M. P. Clark, M. Barlage, B. Rajagopalan, L. Samaniego, G. Abramowitz, and H. Gupta, 2015: Are we unnecessarily constraining the agility of complex process-based models? *Water Resour. Res.*, doi:10.1002/2014WR015820.

Mizukami, N., M. P. Clark, A. J. Newman, A. W. Wood, E. D. Gutmann, B. Nijssen, O. Rakovec, and L. Samaniego, 2017: Towards seamless large-domain parameter estimation for hydrologic models. *Water Resour. Res.*, doi:10.1002/2017WR020401. http://doi.wiley.com/10.1002/2017WR020401 (Accessed September 29, 2017).

Nash, J. E., and J. V Sutcliffe, 1970: River flow forecasting through conceptual models part I — A discussion of principles. *J. Hydrol.*, **10**, 282–290, doi:https://doi.org/10.1016/0022-1694(70)90255-6. http://www.sciencedirect.com/science/article/pii/0022169470902556.

Newman, A., K. Sampson, M. P. Clark, A. R. Bock, R. J. Viger, and D. Blodgett, 2014: A large-sample watershed-scale hydrometeorological dataset for the contiguous USA. doi:doi:10.5065/D6MW2F4D.

Newman, A. J., N. Mizukami, M. P. Clark, A. W. Wood, B. Nijssen, and G. Nearing, 2017: Benchmarking of a Physically Based Hydrologic Model. *J. Hydrometeorol.*, **18**, 2215–2225, doi:10.1175/JHM-D-16-0284.1.

Olden, J. D., and N. L. Poff, 2003: Redundancy and the choice of hydrologic indices for characterizing streamflow regimes. *River Res. Appl.*, doi:10.1002/rra.700.

Oudin, L., V. Andréassian, T. Mathevet, C. Perrin, and C. Michel, 2006: Dynamic averaging of rainfall-runoff model simulations from complementary model parameterizations. *Water Resour. Res.*, **42**, doi:10.1029/2005WR004636. http://doi.wiley.com/10.1029/2005WR004636.

Price, K., S. T. Purucker, S. R. Kraemer, and J. E. Babendreier, 2012: Tradeoffs among watershed model calibration targets for parameter estimation. *Water Resour. Res.*, doi:10.1029/2012WR012005.

Pushpalatha, R., C. Perrin, N. Le Moine, and V. Andréassian, 2012: A review of efficiency criteria suitable for evaluating low-flow simulations. *J. Hydrol.*, **420**–**421**, 171–182, doi:https://doi.org/10.1016/j.jhydrol.2011.11.055. http://www.sciencedirect.com/science/article/pii/S0022169411008407.

Rakovec, O., and Coauthors, 2016a: Multiscale and Multivariate Evaluation of Water Fluxes and States over European River Basins. *J. Hydrometeorol.*, **17**, 287–307, doi:doi:10.1175/JHM-D-15-0054.1. http://journals.ametsoc.org/doi/abs/10.1175/JHM-D-15-0054.1.

Rakovec, O., R. Kumar, S. Attinger, and L. Samaniego, 2016b: Improving the realism of hydrologic model functioning through multivariate parameter estimation. *Water Resour. Res.*, **52**, 7779–7792, doi:10.1002/2016wr019430. http://dx.doi.org/10.1002/2016WR019430.

Samaniego, L., R. Kumar, and S. Attinger, 2010: Multiscale parameter regionalization of a grid-based hydrologic model at the mesoscale. *Water Resour. Res.*, **46**, W05523, doi:10.1029/2008wr007327.

http://dx.doi.org/10.1029/2008WR007327.

Samaniego, L., and Coauthors, 2018: Anthropogenic warming exacerbates European soil moisture droughts. *Nat. Clim. Chang.*, **8**, 421–426, doi:10.1038/s41558-018-0138-5. https://doi.org/10.1038/s41558-018-0138-5.

Seiller, G., R. Roy, and F. Anctil, 2017: Influence of three common calibration metrics on the diagnosis of climate change impacts on water resources. *J. Hydrol.*, doi:10.1016/j.jhydrol.2017.02.004.

Shafii, M., and B. A. Tolson, 2015: Optimizing hydrological consistency by incorporating hydrological signatures into model calibration objectives. *Water Resour. Res.*, **51**, 3796–3814, doi:10.1002/2014wr016520. http://dx.doi.org/10.1002/2014WR016520.

Shamir, E., B. Imam, E. Morin, H. V Gupta, and S. Sorooshian, 2005: The role of hydrograph indices in parameter estimation of rainfall–runoff models. *Hydrol. Process.*, **19**, 2187–2207, doi:10.1002/hyp.5676. https://doi.org/10.1002/hyp.5676.

Thober, S., and Coauthors, 2018: Multi-model ensemble projections of European river floods and high flows at 1.5, 2, and 3 degrees global warming. *Environ. Res. Lett.*, **13**, 14003. http://stacks.iop.org/1748-9326/13/i=1/a=014003.

Tolson, B., and C. Shoemaker, 2007: Dynamically dimensioned search algorithm for computationally efficient watershed model calibration. *Water Resour. Res.*, **43**, doi:10.1029/2005WR004723. https://doi.org/10.1029/2005WR004723.

Westerberg, I. K., and H. K. McMillan, 2015: Uncertainty in hydrological signatures. *Hydrol. Earth Syst. Sci.*, doi:10.5194/hess-19-3951-2015.

——, J. L. Guerrero, P. M. Younger, K. J. Beven, J. Seibert, S. Halldin, J. E. Freer, and C. Y. Xu, 2011: Calibration of hydrological models using flow-duration curves. *Hydrol. Earth Syst. Sci.*, doi:10.5194/hess-15-2205-2011.

Westerberg, I. K., T. Wagener, G. Coxon, H. K. McMillan, A. Castellarin, A. Montanari, and J. Freer, 2016: Uncertainty in hydrological signatures for gauged and ungauged catchments. *Water Resour. Res.*, 1847– 1865, doi:10.1002/2015wr017635. http://dx.doi.org/10.1002/2015WR017635.

Wobus, C., and Coauthors, 2017: Modeled changes in 100 year Flood Risk and Asset Damages within Mapped Floodplains of the Contiguous United States. *Nat. Hazards Earth Syst. Sci.*, **2017**, 1–21, doi:10.5194/nhess-2017-152. https://www.nat-hazards-earth-syst-sci.net/17/2199/2017/nhess-17-2199-2017.html.

Wöhling, T., L. Samaniego, and R. Kumar, 2013: Evaluating multiple performance criteria to calibrate the distributed hydrological model of the upper Neckar catchment. *Environ. Earth Sci.*, **69**, 453–468, doi:10.1007/s12665-013-2306-2. https://doi.org/10.1007/s12665-013-2306-2.

Yadav, M., T. Wagener, and H. Gupta, 2007: Regionalization of constraints on expected watershed response behavior for improved predictions in ungauged basins. *Adv. Water Resour.*, **30**, 1756–1774, doi:http://dx.doi.org/10.1016/j.advwatres.2007.01.005. http://www.sciencedirect.com/science/article/pii/S0309170807000140.

Yilmaz, K. K., H. V Gupta, and T. Wagener, 2008: A process-based diagnostic approach to model evaluation: Application to the NWS distributed hydrologic model. *Water Resour. Res.*, **44**, W09417, doi:10.1029/2007wr006716. http://dx.doi.org/10.1029/2007WR006716.